# Construction of a Two-Dimensional GO/Ti_3_C_2_T_X_ Composite Membrane and Investigation of Mg^2+^/Li^+^ Separation Performance

**DOI:** 10.3390/nano13202777

**Published:** 2023-10-17

**Authors:** Zhenhua Feng, Chengwen Liu, Binbin Tang, Xiaojun Yang, Wenjie Jiang, Peng Wang, Xianjun Tang, Hongshan Wang, Xiangdong Zeng, Guangyong Zeng

**Affiliations:** 1Evaluation and Utilization of Strategic Rare Metals and Rare Earth Resource Key Laboratory of Sichuan Province, Chengdu Mineral Resources Supervision and Testing Center, Ministry of Land and Resources, Chengdu 610081, China; fengzhenhua@alu.cdut.edu.cn (Z.F.); tangbinbin2013@alu.cdut.edu.cn (B.T.); jwenjie1988@sina.com (W.J.); 2Chengdu Analytical & Testing Center for Mineral and Rocks, Sichuan Bureau of Geology and Mineral Resources, Chengdu 610081, China; 3College of Materials and Chemistry & Chemical Engineering, Chengdu University of Technology, Chengdu 610059, China; chengwenliu@stu.cdut.edu.cn (C.L.); hongshanwang@stu.cdut.edu.cn (H.W.); zengxiandong17@cdut.edu.cn (X.Z.); 4Sichuan Salt Geology Drilling Team (Sichuan Mineral Salt Mining Engineering Technology Research Center), Zigong 643000, China; wpeng1818@google.com (P.W.); txjly1988@sina.com (X.T.)

**Keywords:** graphene oxide, Ti_3_C_2_T_X_ nanosheets, two-dimensional materials membrane, Mg^2+^/Li^+^ separation

## Abstract

Graphene oxide (GO) two-dimensional (2D) membranes with unique layer structures and tunable layer spacing have special advantages and great potential in the field of water treatment. However, GO membranes face the issues of weak anti-swelling ability as well as poor permeability. We prepared GO/Ti_3_C_2_T_X_ 2D composite membranes with 2D/2D structures by intercalating Ti_3_C_2_T_X_ nanosheets with slightly smaller sizes into GO membranes. Ti_3_C_2_T_X_ intercalation can effectively expand the layer spacing of GO, thereby substantially enhancing the flux of the composite membrane (2.82 to 6.35 L·m^−2^·h^−1^). Moreover, the GO/Ti_3_C_2_T_X_ composite membrane exhibited a good Mg^2+^/Li^+^ separation capability. For the simulated brine, the separation factor of M2 was 3.81, and the salt solution flux was as high as 5.26 L·m^−2^·h^−1^. Meanwhile, the incorporation of Ti_3_C_2_T_X_ nanosheets significantly improved the stability of GO/Ti_3_C_2_T_X_ membranes in different pH environments. This study provides a unique insight into the preparation of highly permeable and ion-selective GO membranes.

## 1. Introduction

With the vigorous growth of the new energy industry in recent years, lithium-ion batteries have become an industry hot spot [1,2,3]. This is leading to an increase in demand for lithium resources worldwide. However, the cost of extracting lithium from lithium ores using traditional methods is too high, and the environmental pollution is more serious. Therefore, the exploitation of lithium in liquid mines such as salt lakes has become one of the most effective ways to alleviate the current shortage of lithium resources [4,5]. Traditional lithium extraction techniques from salt lakes include precipitation, ion exchange, adsorption, and extraction. However, the above process has shortcomings such as high energy consumption, complex operation, and environmental problems [6,7]. Hence, the development of an efficient and green method for lithium extraction is of great significance.

Membrane technology is a separation and purification method with a simple process, low energy consumption, less use of chemical reagents, and a green environment [8,9]. There are more and more studies showing that membrane technology can be used for the extraction of lithium from lithium-containing solutions [10]. However, the membrane still faces problems such as low permeability, easy pollution of the membrane surface, and poor selectivity of lithium ions [3,11,12]. Common commercial membranes are dominated by inorganic material membranes and organic polymer membranes. The complexity of the preparation process of inorganic membranes leads to their high cost, while the poor mechanical strength of polymer membranes and the serious problem of membrane contamination lead to a short service life [13]. The trade-off effect of membranes restricts the improvement of their separation efficiency. In recent years, emerging 2D materials have provided new directions for the selection of high-performance membrane materials [14]. Two-dimensional materials represented by graphene oxide (GO), MoS_2_, and metal-organic frameworks can be stacked in a layer-by-layer fashion to form 2D layered membranes with nano/sub-nano channels [15]. GO is a typical two-dimensional layered nanosheet containing many epoxy, carboxyl, and hydroxyl groups on the surface and edge portions of GO nanosheets [16]. This enables GO to have excellent hydrophilicity and chemical modification properties. These functional groups can provide binding sites for the targets during the separation process, making GO membranes possess a good separation effect in the case of combined-size screening [17,18]. However, the functional groups of GO tend to generate hydrogen bonds between water molecules, thus increasing the osmotic resistance [19,20]. Moreover, the distance between layers of GO nanosheets is small (∼8 Å), so the permeability of membranes based only on GO is poor [21,22]. To solve this challenge, researchers have realized the enhancement of GO membrane permeation performance by dopamine-modified GO [23], MoS_2_ nanosheet doping [19], and chitosan coating [24]. In addition, GO nanosheets are prone to interact with water molecules in aqueous systems, leading to the swelling of GO membranes [25,26,27,28,29]. This is a difficult problem, limiting the application of GO membranes. Xi et al. [30] reduced GO to rGO, thus reducing the oxygen-containing functional groups of GO membranes. This resulted in a greatly enhanced stability of GO membranes doped with partially rGO nanosheets.

MXene is a general term for transition metal carbon/nitrides that exhibit a 2D nanosheet structure. It is characterized by large lateral size and ultra-thin thickness (the atomic level), a high specific surface area, and abundant hydrophilic groups (-F -O, -OH) [31,32]. Therefore, MXene has a wide range of applications in the fields of energy [33,34], catalysis, electrochemistry [35], environment [36,37], and biology [38]. The general formula of MXene can be expressed as M_n+1_X_n_T_x_, which can be prepared by etching the precursor MAX phase (M_n+1_AX_n_). M represents a transition metal element, A is aluminum or silicon, and X represents a carbon or nitrogen element. T_x_ is a surface functional group of MXene nanosheets (n = 1, 2, 3) [39,40]. The existing studies have shown that 2D membranes obtained by stacking MXene nanosheets have good permeability. MXene composite membranes have unique two-dimensional channels, excellent hydrophilicity, and chemical modifiability. They are widely used in oil-water separation [41], dye wastewater treatment [42], seawater desalination [43,44], and pharmaceutical wastewater treatment [45]. Ren et al. [46] constructed 2D MXene membranes on PVDF substrate membranes using vacuum filtration. The water flux was as high as 37.4 L·m^−2^·h^−1^·bar^−1^ and exhibited good selectivity for MB, Al^3+^, Mg^2+^, and Ca^2+^.

In this work, we obtained GO/Ti_3_C_2_T_X_ membranes prepared by pressure-assisted filtration on the Polyethersulfone (PES) substrate. Owing to the small size of MXene nanosheets, which can be inserted into the interlayer of GO, Ti_3_C_2_T_X_ nanosheets can be combined with GO through van der Waals force interaction. Moreover, Ti_3_C_2_T_X_ can compensate for the non-selective defects caused by the stacking of GO layers, thus extending the path of ions in the interlayer nano-channels (improving the ion rejection of the composite membrane). The characterization of GO/Ti_3_C_2_T_X_ composite membranes showed that Ti_3_C_2_T_X_ was uniformly dispersed inside the composite membranes, and the GO/Ti_3_C_2_T_X_ membranes still retained the lamellar structure. The layer spacing of GO/Ti_3_C_2_T_X_ membranes was increased, which effectively increased the pure water flux of GO/Ti_3_C_2_T_X_ membranes. In addition, GO/Ti_3_C_2_T_X_ composite membranes presented high Mg^2+^ rejection and excellent Li^+^ permeability. The GO/Ti_3_C_2_T_X_ membranes showed good Mg^2+^/Li^+^ selectivity for a mixed solution of MgCl_2_ and LiCl (Mg^2+^/Li^+^ mass ratio = 20). In addition, the introduction of Ti_3_C_2_T_X_ nanosheets decreased the repulsion between the GO layers. Hence, the GO/Ti_3_C_2_T_X_ membrane possesses better stability in water than the pure GO membrane.

## 2. Materials and Methods

### 2.1. Materials

Ti_3_AlC_2_ was purchased from Jilin 11 Technology Co. (Jilin, China). LiF, HCl, and NaNO_3_ were provided by Aladdin (Shanghai, China). KMnO_4_ and anhydrous MgCl_2_ were purchased from Chengdu Kelong (Chengdu, China). Anhydrous LiCl, H_2_SO_4_, and H_2_O_2_ were provided by Shanghai Adamas Reagent (Shanghai, China). PES microfiltration membranes (0.22 µm) were purchased from Tianjin Jinteng (Tianjin, China), and NaOH was purchased from Macklin Reagent (Shanghai, China).

### 2.2. Fabrication of GO

GO was prepared by the modified Hummers’ method [47]. Specifically, NaNO_3_ and H_2_SO_4_ were mixed in a beaker, and then a certain amount of natural graphite powder was added and stirred to obtain pre-oxidized graphite (2.5 h). Subsequently, KMnO_4_ was gradually incorporated into the above solution and magnetically stirred for 1 h. A H_2_O_2_ solution was added for further oxidation. The mixture was then centrifuged (3500 rpm) and washed with DI water until the solution had a pH = 7 to remove impurities and residual acid. Finally, GO nanosheets were obtained by ultrasonically peeling off multilayers of GO, and freeze-drying was used to obtain GO powder.

### 2.3. Preparation of Ti_3_C_2_T_X_ Nanosheets

2D Ti_3_C_2_T_X_ was fabricated by chemical etching the MAX phase using ultrasound-assisted stripping with a mixed solution of LiF + HCl [48,49]. First, 50 mL of HCl (9 M) was added to a polytetrafluoroethylene beaker, followed by the incorporation of 0.6 g of LiF into the HCl solution, stirring continuously until completely dissolved. Then, 0.5 g of Ti_3_AlC_2_ was added to the beaker and magnetically stirred at 30 °C for 28 h. The unstripped precipitate was separated from the supernatant by means of centrifugation (5000 rpm) and then washed several times repeatedly using deionized (DI) water until the supernatant pH > 6. The multilayered Ti_3_C_2_T_X_ powder was obtained after drying at 50 °C. Subsequently, the multilayered Ti_3_C_2_T_X_ nanosheets were added to DI water and ultrasonically stripped under a nitrogen atmosphere for 3 h to obtain monolayered Ti_3_C_2_T_X_ nanosheets (25 °C). Finally, the multilayer Ti_3_C_2_T_X_ was removed by centrifugation at 5000 rpm, and the supernatant was collected and freeze-dried for 30 h to obtain monolayer Ti_3_C_2_T_X_ nanosheets.

### 2.4. Construction of GO/Ti_3_C_2_T_X_ Composite Membranes

As shown in Figure 1, the GO nanosheets were first dispersed in 50 mL of DI water and ultrasonic for 20 min to obtain a homogeneous dispersion; in addition, the Ti_3_C_2_T_X_ powder was added to another beaker containing 50 mL of DI water and ultrasonic for 15 min to make it well-dispersed. Then, the GO dispersion was mixed with the Ti_3_C_2_T_X_ dispersion, and the GO/Ti_3_C_2_T_X_ dispersion was obtained by sonication for another 15 min. The above dispersion was filtered onto a PES microfiltration membrane through a dead-end filtration device, and the GO/Ti_3_C_2_T_X_ composite membrane was finally obtained. The ratio of GO to Ti_3_C_2_T_X_ in the composite membrane is shown in Table 1.

### 2.5. Characterization of Membranes

The crystal structures of MAX, Ti_3_C_2_T_X_, and GO powders, as well as the interlayer spacing variation of GO/Ti_3_C_2_T_X_ membranes, were characterized by X-ray diffraction (XRD). The elemental content of the composite membrane surface was tested by X-ray photoelectron spectroscopy (XPS). Scanning electron microscopy (SEM) was used to investigate the microscopic morphology of GO, MAX phase, and Ti_3_C_2_T_X_, as well as the GO/Ti_3_C_2_T_X_ membranes. In addition, the hydrophilicity of the membrane surface was evaluated by means of a contact angle meter, and the contact angle was obtained by averaging three tests per membrane.

### 2.6. Membrane Performance Testing

The permeability and Mg^2+^/Li^+^ separation ability of the composite membranes were tested using a laboratory-constructed dead-end filtration unit (effective area of 12.56 cm^−2^). First, all membranes were pre-pressurized with DI water at 2.5 bar for 30 min. The flux (*F*: L·m^−2^·h^−1^) of the membrane was tested at 2 bar, and the volume of deionized water on the permeate side was collected and tested by Equation (1):(1)$$F=\frac{V}{A\times t}$$
where *V* (L) represents the permeate volume, *A* is the permeate area (m^2^), and *t* (h) represents the tested time for permeation.

Tested with 1 g/L LiCl and MgCl_2_ solutions to show the separation ability of membranes, respectively, and the rejection (*R*%) was tested by Equation (2):(2)$$R(\%)=\left(1-\frac{C_{p}}{C_{f}}\right)\times 100\%$$
where *C_f_* is the ion concentration on the feed side and *C_p_* is the ion concentration in the permeate. A conductivity meter is used to test the conductivity of the solution before and after permeation to derive the salt concentration (DDS-307A).

The Mg^2+^/Li^+^ selectivity of the composite membrane was tested using a mixed solution of MgCl_2_ and LiCl (2 g/L), where the Mg^2+^/Li^+^ mass ratio was 20. The Mg^2+^/Li^+^ selectivity of the composite membrane was evaluated by the separation factor *S_Li,Mg_*, as in Equation (3):(3)$$S_{Li,Mg}=\frac{C_{Li,p}/C_{Mg,p}}{C_{Li,f}/C_{Mg,f}}$$
where *C_Li,p_* is the concentration of Li^+^ in the permeate solution and *C_Mg,p_* is Mg^2+^ on the permeate solution; *C_Li,f_* is the concentration of Li^+^ in the feed solution and *C_Mg,f_* is Mg^2+^ on the feed side. Inductively coupled plasma optical emission spectroscopy (ICP-OES) was used to test the concentration of each ion in the salt solution before and after the permeate process.

## 3. Results and Discussion

### 3.1. Characterization of GO and Ti_3_C_2_T_X_ Nanosheets

Figure 2a shows the XRD spectra of GO, Ti_3_AlC_2_, and Ti_3_C_2_T_x_. A narrower peak (the characteristic peak of GO nanosheets) was presented at 2θ = 10.5° (001), which proved the successful synthesis of GO. The XRD results of Ti_3_C_2_T_x_ showed that the intensity of the 104 peak was much weakened compared to that of Ti_3_AlC_2_, which implied that the Al atomic layer of Ti_3_AlC_2_ was successfully etched. Moreover, the 002 peak of Ti_3_C_2_T_x_ shifted from 9.5° to 6.1° for Ti_3_AlC_2_, which indicated an increase in its interlayer spacing. All the above results demonstrated the successful synthesis of GO and Ti_3_C_2_T_X_ nanosheets. Figure 2b reveals the morphology of tightly stacked GO nanosheets. The MAX phase (Figure 2c) is a larger-sized bulk structure with tighter layers. The Ti_3_C_2_T_X_ nanosheets obtained by etching are sparser and show a clear layer structure (Figure 2d).

Figure 3a,b shows the surface of the original GO membrane (M0) at different multiplicities. M0 exhibited a large number of wrinkled structures, which are due to the flexible GO nanosheets with larger sizes when stacked [22]. And its surface is relatively smoother, without obvious defects. Observation of the cross-section of the M0 indicated that the GO nanosheets were arranged in a very regular manner. The M0 exhibited a typical 2D layered structure, with layers stacked densely. It is the nano-restricted domain channels created by the stacking of GO nanosheets on top of each other that allow GO membranes to trap targets larger than their layer spacing (Figure 3c,d). After intercalation into the GO membrane by Ti_3_C_2_T_X_ nanosheets, the surface of the M2 was rougher, and the appearance of lamellar Ti_3_C_2_T_X_ on its surface can be observed. However, the M2 still showed more folds, which indicated that its overall structure had not been changed (Figure 3e,f). The cross-sectional area of the M2 showed an increase in the thickness of their 2D-separated layers, and their lamellar structure was sparser than that of the M0. In addition, Ti_3_C_2_T_X_ nanosheets with smaller sizes can be observed between the layers, which indicates the successful intercalation of Ti_3_C_2_T_X_ (Figure 3g,h).

Figure 4 shows the EDS-mapping image of the M2, and it can be clearly observed that the Ti and F elements were more evenly distributed on the top of the membrane. This demonstrated that the Ti_3_C_2_T_X_ nanosheets were uniformly dispersed in the GO membrane.

In the XPS spectra of M0, the fitted peaks of C 1s and O 1s appeared at 284.8 eV and 530 eV, which was consistent with the literature reports [50,51]. After the addition of Ti_3_C_2_T_X_, the fitted peaks at 682.4 eV and 457 eV corresponded to F 1s and Ti 2p, respectively (Figure 5a) [52]. The above results demonstrated the successful introduction of Ti_3_C_2_T_X_ nanosheets into GO. The membrane surfaces of M0 and M2 were further analyzed using high-resolution C 1 spectroscopy. The 284.8 eV, 286.9 eV, and 288.5 eV in Figure 5b correspond to the C-C/C=C, C-O, and C=O/COOH present in GO nanosheets [53], respectively (Figure 5b). The C 1s spectrum of the M2 (Figure 5c) showed an additional fitted peak at 282.3 eV, which corresponds to the C-Ti in the Ti_3_C_2_T_X_ nanosheets [29]. The convolution peaks appearing at 455.8, 458.7, 460.2, 461.6, and 464.4 eV correspond to C-Ti-(O, OH), Ti_x_-O_y_, TiO_2_, Ti-C_x_, and C-Ti-F (Figure 5d) [54]. The surface element contents of M0 and M2 are shown in Table 2. The addition of F and Ti elements in M2 proved that Ti_3_C_2_T_X_ was successfully doped in the GO layer.

Figure 6 shows the XRD patterns of M0 and M2. For the M0, the characteristic peak of GO nanosheets was located at 2θ = 11.1°. The layer spacing d = 7.94 Å of the M0 at this time was calculated by Bragg’s equation. After the intercalation by Ti_3_C_2_T_X_, the characteristic peak of M2 was slightly shifted to the left at 2θ = 10.4°, and the layer spacing d = 8.5 Å of the M2. The above results proved that the 2D Ti_3_C_2_T_X_ nanosheets could enter the interlayer of GO, which led to the widening of layer spacing in the GO 2D membranes. This will reduce the permeation resistance of water molecules and effectively enhance the water flux [55].

### 3.2. The Performance of GO/Ti_3_C_2_T_X_ Composite Membrane

Figure 7a shows the water contact angle (WCA) of different membrane surfaces. According to reports [56], the lower the water contact angle on the membrane surface, the better hydrophilicity the membrane possesses. The CA of the original M0 was 39°, indicating that the GO membrane was more hydrophilic. Since GO nanosheets contain a large number of hydrophilic groups such as hydroxyl, carboxyl, and epoxy groups, these groups can have hydrogen bonding interactions with water molecules, thus allowing the membrane surface to absorb more water molecules and accelerating the formation of the hydration layer. After adding Ti_3_C_2_T_X_ to the GO membrane, the CA of all GO/Ti_3_C_2_T_X_ membranes was higher than M0. Since the surface hydrophilic functional groups of Ti_3_C_2_T_X_ nanosheets were less than those of GO, the hydrophilicity of GO/Ti_3_C_2_T_X_ membranes was weaker compared with M0.

Figure 7b shows the water flux of the original GO membranes loaded with different contents. The pure water flux of the GO membrane gradually decreased with the increase in GO content. The GO membrane loaded with 2 mg showed the lowest water flux (2.27 L·m^−2^·h^−1^), and due to the small layer spacing of the 2D GO membrane, its permeability is usually poor compared to other 2D membranes (such as 2D Ti_3_C_2_T_X_ membranes and MoS_2_ membranes). In addition, the increase in GO loading significantly increased the thickness of the separation layer, leading to enhanced permeation resistance. Hence, the permeability of the membrane gradually decreased. In addition, the separation ability of GO membranes with different contents was tested using 1 g/L MgCl_2_. From Figure 7c, it can be observed that the flux of the MgCl_2_ salt solution showed a decreasing trend as the GO loading increased, which was similar to its pure water flux changes (11 to 1.93 L·m^−2^·h^−1^). The retention of MgCl_2_ by the GO membrane was enhanced with increasing GO content (42.1% to 63.3%). Since the hydration diameter of Mg^2+^ is 8.56 Å, it is slightly larger than the layer spacing of the GO membrane (7.94 Å). Under the action of size screening [57], the larger-sized Mg^2+^ is intercepted by the GO membrane, which achieves better Mg^2+^ retention. However, the results showed that the permeability of GO membranes was bound to be poor if they were to realize a good ion retention effect. This indicated that the separation effect of the GO membrane was always constrained by the trade-off effect.

After mixing Ti_3_C_2_T_X_ nanosheets in 2D GO-based membranes, the flux of different ratios of GO/Ti_3_C_2_T_X_ membranes was enhanced (Figure 7a). It increased from 2.8 L·m^−2^·h^−1^ in M0 to the highest 13.2 L·m^−2^·h^−1^ in M4. This was attributed to the fact that the intercalation of Ti_3_C_2_T_X_ nanosheets effectively enhanced the layer spacing of the GO membrane. The transport resistance of water molecules was reduced, resulting in a substantial increase in the permeability of the GO/Ti_3_C_2_T_X_ membrane. To evaluate the separation effect of GO/Ti_3_C_2_T_X_ membranes, a 1 g/L MgCl_2_ solution was used for testing. As shown in Figure 7d, the Mg^2+^ rejection of M1 was similar to that of M0. Since the layer spacing of the GO/Ti_3_C_2_T_X_ membrane has not changed significantly currently, the rejection of Mg^2+^ by M2 was increased to 65.2%, and the flux was twice that of M0. The enhancement of Mg^2+^ rejection was mainly due to the addition of Ti_3_C_2_T_X_ to increase the thickness of the membrane, while the separation capacity of M2 was enhanced by the synergistic effect of size-sieving action and adsorption [58]. With the excessive addition of Ti_3_C_2_T_X_, the MgCl_2_ flux of M3 and M4 increased greatly, but the rejection of Mg^2+^ also showed a significant decline. This was because too many Ti_3_C_2_T_X_ nanosheets may cause the layer spacing of the GO/Ti_3_C_2_T_X_ membrane to be larger than the hydration diameter of Mg^2+^. In addition, all GO/Ti_3_C_2_T_X_ membranes showed low rejection of LiCl (Figure 7e). This was attributed to Li^+^ having a smaller hydration diameter (7.64 Å) [59]. The permeability of the different membranes was combined, as was the difference in rejection for Mg^2+^ and Li^+^. M2 was selected as the best membrane to investigate its Mg^2+^/Li^+^ separation ability.

A mixed solution of MgCl_2_ and LiCl (Mg^2+^/Li^+^ mass ratio of 20, 2 g/L) was used as a simulated brine. As shown in Figure 7f, the flux of the salt solution of M0 was 2.5 L·m^−2^·h^−1^, and the S_Li,Mg_ was 3.35. The permeability of GO/Ti_3_C_2_T_X_ membranes with different ratios was enhanced by Ti_3_C_2_T_X_ intercalation modification. M2 had the highest separation ability (S_Li,Mg_ = 3.81), while the permeability (5.26 L·m^−2^·h^−1^) was enhanced by ~100% compared to M0. With the further addition of Ti_3_C_2_T_X_, the S_Li,Mg_ of both M3 and M4 were significantly lower, despite the substantial increase in permeability. The M2 had a suitable layer spacing (8.5 Å) after being modified with Ti_3_C_2_T_X_. Compared to the original M0, the increased layer spacing of M2 can significantly reduce the permeation resistance of water molecules and increase the flux of the membrane. In addition, the layer spacing of M2 was slightly smaller than the hydration diameter of Mg^2+^. This made M2 possess better Mg^2+^ rejection. On the other hand, the layer spacing of M2 is larger than the hydration diameter of Li^+^, so M2 has excellent permeability to Li^+^. In summary, M2 possessed good permeability and Mg^2+^/Li^+^ separation ability.

Since 2D GO membranes are highly susceptible to swelling in water, this can significantly reduce the service life of the membranes [60]. It was tested for the stability of M0 and M2 to evaluate their anti-swelling ability under different conditions of use (Figure 8). M0 and M2 were immersed in the corresponding acidic, neutral, and alkaline solutions at pH = 3, 7, and 11. It was clearly found that the surface of the M0 gradually changed from yellow to brown with the increase in immersion time. The separation layer under the three different pHs occurred with different degrees of detachment and disintegration. For M2, the membrane surface color did not change after 14 days of immersion in different acidic and alkaline conditions. The surface separation layer was structurally complete and did not produce peeling or cracking. The above experimental results indicated that the doping of Ti_3_C_2_T_X_ nanosheets will effectively improve the anti-swelling ability of the GO/Ti_3_C_2_T_X_ membrane, thus enhancing its stability in water. Since Ti_3_C_2_T_X_ nanosheets weaken the mutual repulsion between the GO layers, the separate layer is more stable [51,61].

## 4. Conclusions

In this work, a novel two-dimensional GO/Ti_3_C_2_T_X_ composite membrane was prepared by pressure-assisted filtration and investigated for its Mg^2+^/Li^+^ separation performance. SEM, XRD, and XPS were used to characterize the microscopic morphology and physicochemical properties of the GO/Ti_3_C_2_T_X_ membrane and to demonstrate the successful intercalation of Ti_3_C_2_T_X_ nanosheets into the GO membrane. The permeation performance of the optimal GO/Ti_3_C_2_T_X_ membrane (M2) was significantly improved (from 2.82 to 6.35 L·m^−2^·h^−1^) compared to the original GO membrane, which is almost three times better. Ti_3_C_2_T_X_ nanosheets functioned as an intercalation to expand the layer spacing in GO membranes, which made the two-dimensional layered structure of the composite membranes sparser. Thus, the water permeability of the GO/Ti_3_C_2_T_X_ membrane was increased. In addition, the enhanced layer spacing (8.5 Å) of GO/Ti_3_C_2_T_X_ membranes was still smaller than the hydration diameter of Mg^2+^. The size-sieving effect made the composite membrane possess a good Mg^2+^ rejection and a very low Li^+^ rejection. This resulted in a good Mg^2+^/Li^+^ selective separation (S_Li,Mg_ = 3.8). The GO/Ti_3_C_2_T_X_ membrane also exhibited excellent anti-swelling ability and was more stable than the GO membrane under different acid-base conditions. This 2D/2D combination provides a new idea for the further development of 2D GO membranes in the field of Mg^2+^/Li^+^ separation.

## Figures and Tables

**Figure 1 nanomaterials-13-02777-f001:**
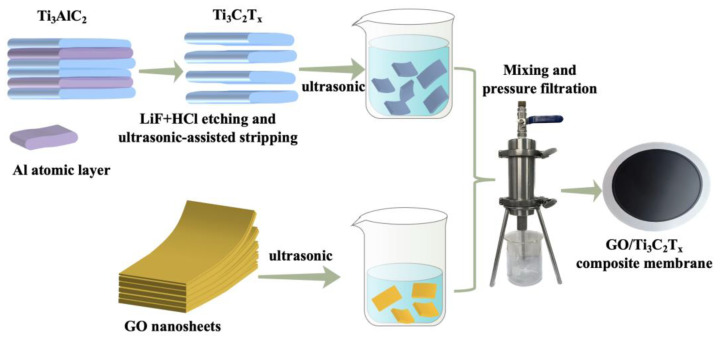
Schematic diagram of GO/Ti_3_C_2_T_X_ composite membrane preparation.

**Figure 2 nanomaterials-13-02777-f002:**
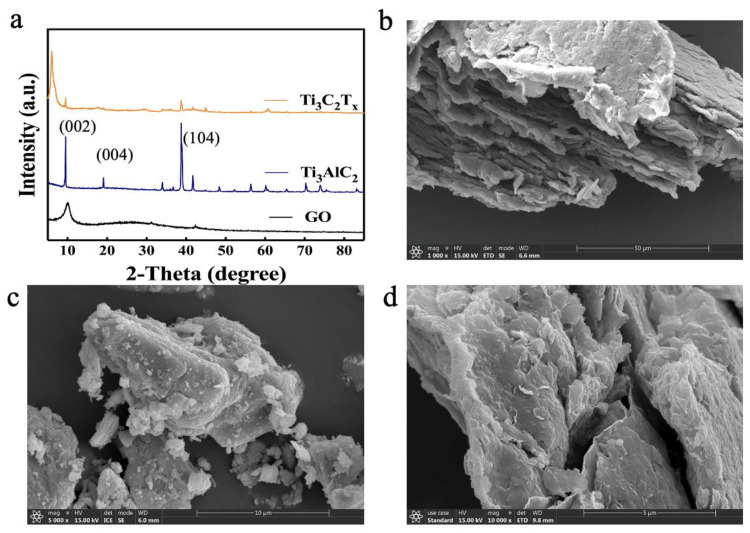
(**a**) XRD pattern of MAX and Ti_3_C_2_T_X_; SEM images of (**b**) GO, (**c**) MAX, and (**d**) Ti_3_C_2_T_X_.

**Figure 3 nanomaterials-13-02777-f003:**
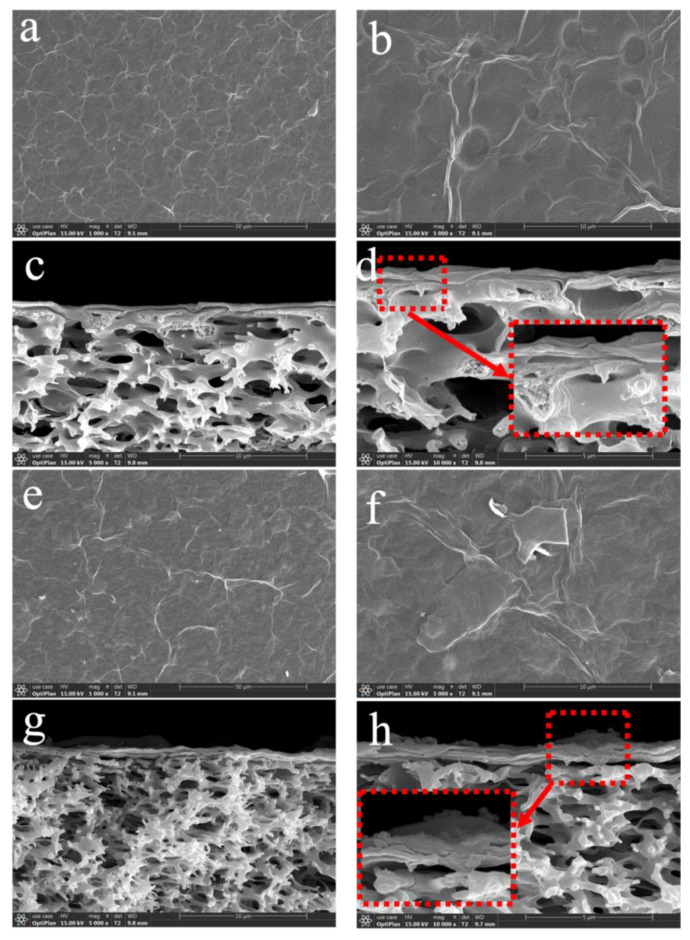
(**a**,**b**,**e**,**f**) surface and (**c**,**d**,**g**,**h**) cross-section of SEM images for M0 and M2.

**Figure 4 nanomaterials-13-02777-f004:**
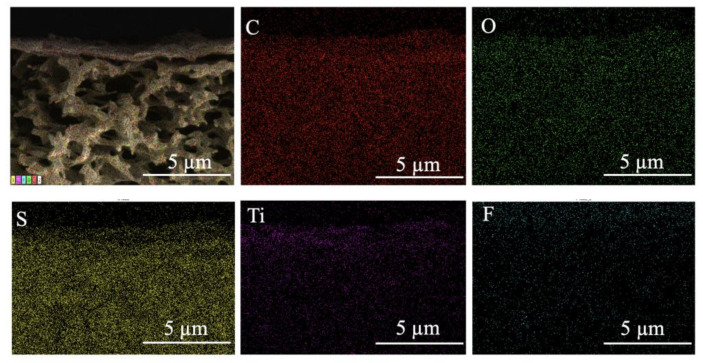
EDS mapping of M2.

**Figure 5 nanomaterials-13-02777-f005:**
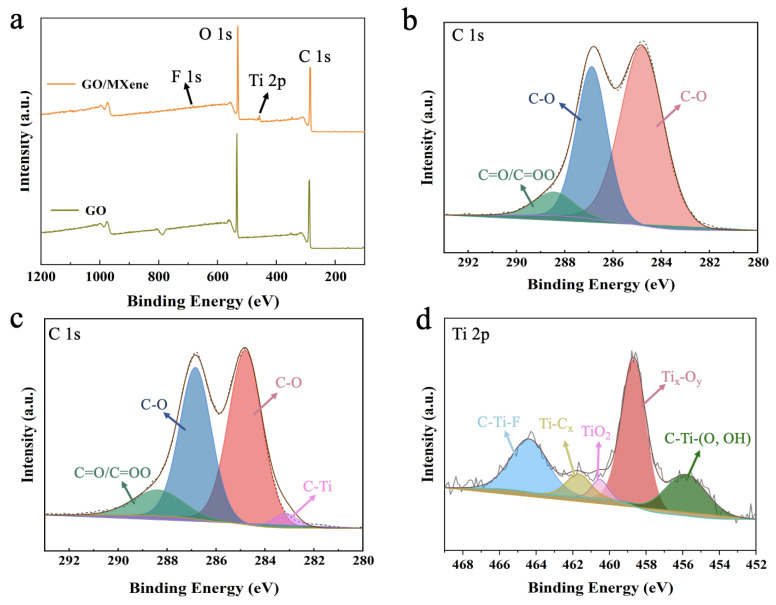
(**a**) XPS full spectra of M0 and M2; (**b**,**c**) High-resolution C 1s spectra of M0 and M2; (**d**) Ti 2p spectra of M2.

**Figure 6 nanomaterials-13-02777-f006:**
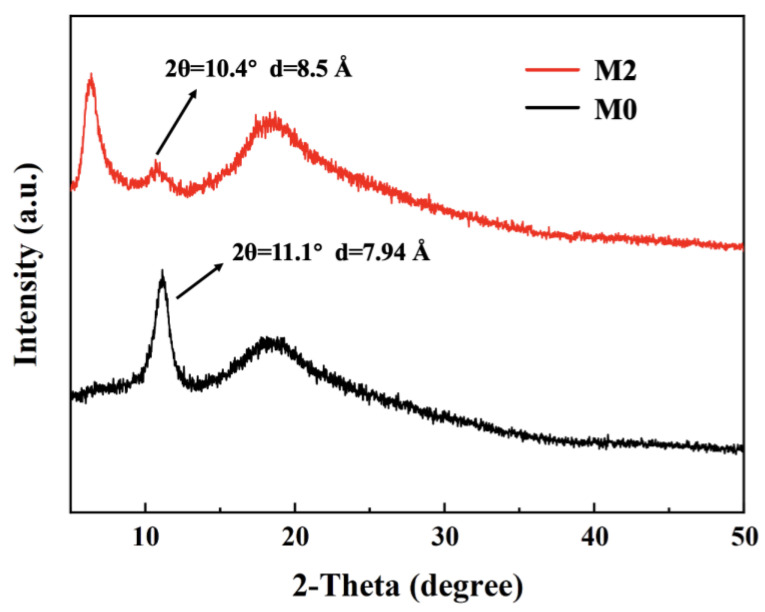
XRD spectra of M0 and M2.

**Figure 7 nanomaterials-13-02777-f007:**
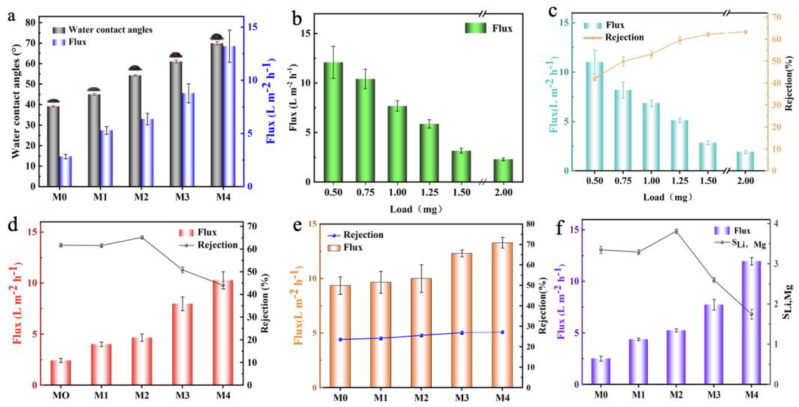
(**a**) CA and water flux of GO/Ti_3_C_2_T_X_ membranes; (**b**,**c**) Water flux and MgCl_2_ separation ability with different content of GO membranes; (**d**,**e**) Separation performance of GO/Ti_3_C_2_T_X_ membranes for MgCl_2_ and LiCl solutions; (**f**) Mg^2+^/Li^+^ separation ability of GO/Ti_3_C_2_T_X_ membranes.

**Figure 8 nanomaterials-13-02777-f008:**
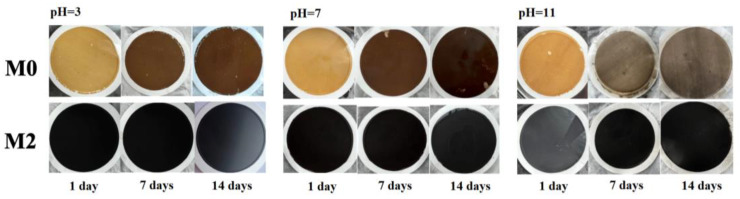
Stability test of M0 and M2 under different conditions.

**Table 1 nanomaterials-13-02777-t001:** Composition of different GO/Ti_3_C_2_T_X_ composite membranes.

Membrane	GO (mg)	Ti_3_C_2_T_X_ (mg)
M0	1.5	0
M1	1.5	0.5
M2	1.5	1
M3	1.5	1.5
M4	1.5	2

**Table 2 nanomaterials-13-02777-t002:** XPS analysis results shows elemental content on the M0 and M2 surfaces.

Membrane	C (%)	O (%)	Ti (%)	F (%)
M0	72.1	27.9	0	0
M2	69.74	28.24	0.97	1.05

## Data Availability

The data that supports the findings of this study is available.

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
