# Peer review of "Construction of a Two-Dimensional GO/Ti3C2TX Composite Membrane and Investigation of Mg2+/Li+ Separation Performance"

_nanomaterials, 2023, doi:10.3390/nano13202777_

Round 1

Reviewer 1 Report

In this article, Z Feng et.al, demonstrated the study of Mg2+/Li+ separation performances using 2D GO/MXene composite membrane.  The manuscript was well written and the concept theme is good. The results presented in the study and the characterization reports are well in accordance, which could be likely to attract the readers in Mg2+/Li+ separation using membrane applications However, there are some issues that needs to be clarified before processing for the further step toward publication, and here are some of my concerns as given below.

1)      Title: (1) Ti3C2 is only one type of MXene and cannot be represented by MXene (the author only studied Ti3C2). (2) Ti3C2? or Ti3C2Tx? Please Rename the title.

2)      Authors have clearly portraited their view of the implications of GO and the solutions. However, it is not clear that how and why the GO needs to combine with MXene layer and how does it could be different from the other suggested materials like MoS2 doping, dopamine etc. Please emphasize the importance of GO-MXene composite. Since GO and MXene are both belongs 2D materials, so please use their synergistic effect for your studies, should be briefly explain in the introduction.

3)      Importance MXene should further be enhanced by their wide applicability in various field and how it could be useful in GO/MXene membranes Mg2+/Li+ selectivity and separation process.

4)      Please use the full abbreviation if the authors use any word for the first time in the manuscript, for example PES substrate.

5)      In Fig.1 Please indicate the MAX formula and the color of Al layer (Violet) which was removed after the etching process.

6)      Please indicate the optimized GO/MXene sample (M2) in the characterizations, its looks confusing that which GO/MXene composite is optimized. Also provide the elemental percentage of composite membranes. Please unify all the names indexed in side the Figures, for the better understanding to readers.

7)      Please provide the BET surface area and pore size analysis for the better comparison of the surface area of the composite’s samples.

8)      Please study some electrical properties (electrical conductivity or impedance) of the composite materials by comparing with the pristine counterparts for exploring the fact of the synergistic effects of GO/MXene. This could further helpful to the readers to understand the transport resistance of water.

9)      Introduction: (1) Why study MXene based composite membranes, and their wide application in various filed and how does it helpful to enhance the separation process in in membranes. Some of the latest reports are suggested to cite at the MXene related discussions, such as Rare Metals 202140 (6), 1459– 1476, Desalination 537 (2022): 115847. ACS Materials Letters 5 (2023): 2739-2746.

Author Response

For detailed point-by-point response to the reviewer’s comments, please download the word document shown below.

Reviewer 2 Report

In this work the authors have prepared graphene oxide (GO) two-dimensional (2D) membranes intercalated with MXene 20 nanosheets. The resulting GO/MXene 2D composite membranes were tested for the separation of Mg2+/Li+ mixtures from synthetic brines. The composite membrane with a certain loading of the MXene showed an increased permeability and a slightly better Mg2+/Li+ separation than the pristine GO membrane. Also, the composite membrane appeared to be more stable than the original GO membrane. This paper contains some interesting data and could potentially become publication-worthy upon further revision. Specifically:

1.    Have the authors prepared any pure MXene membranes? How is the performance of such membranes in treating the same synthetic brine solutions?

2.    The resolution of the SEM images is not sufficient to validate the layered structure of these materials or the claim that MXene single sheets incorporate within the GO structure. Higher resolution SEM would be of value.

3.    The quality of the technical English and graphics needs to be improved,     

Needs to be improved. 

Author Response

(The authors gave the same response as above.)

Round 2

Reviewer 2 Report

I am fine with the authors response and revision of their paper.

Needs to be improved.